# The Extracellular Bone Marrow Microenvironment—A Proteomic Comparison of Constitutive Protein Release by In Vitro Cultured Osteoblasts and Mesenchymal Stem Cells

**DOI:** 10.3390/cancers13010062

**Published:** 2020-12-28

**Authors:** Elise Aasebø, Even Birkeland, Frode Selheim, Frode Berven, Annette K. Brenner, Øystein Bruserud

**Affiliations:** 1Department of Clinical Science, University of Bergen, N-5021 Bergen, Norway; elise.aasebo@uib.no (E.A.); annette.brenner@uib.no (A.K.B.); 2The Proteomics Facility of the University of Bergen (PROBE), University of Bergen, N-5021 Bergen, Norway; even.birkeland@uib.no (E.B.); frode.selheim@uib.no (F.S.); frode.berven@uib.no (F.B.); 3Department of Medicine, Haukeland University Hospital, N-5021 Bergen, Norway

**Keywords:** cancer, bone marrow, osteoblasts, mesenchymal stem cells, protein, proteomics, conditioned medium, extracellular matrix, exosome

## Abstract

**Simple Summary:**

Normal blood cells are formed in the bone marrow by a process called hematopoiesis. This process is supported by a network of non-hematopoietic cells including connective tissue cells, blood vessel cells and bone-forming cells. However, these cells can also support the growth of cancer cells, i.e., hematological malignancies (e.g., leukemias) and cancers that arise in another organ and spread to the bone marrow. Two of these cancer-supporting normal cells are bone-forming osteoblasts and a subset of connective tissue cells called mesenchymal stem cells. One mechanism for their cancer support is the release of proteins that support cancer cell proliferation and progression of the cancer disease. Our present study shows that both these normal cells release a wide range of proteins that support cancer cells, and inhibition of this protein-mediated cancer support may become a new strategy for cancer treatment.

**Abstract:**

Mesenchymal stem cells (MSCs) and osteoblasts are bone marrow stromal cells that contribute to the formation of stem cell niches and support normal hematopoiesis, leukemogenesis and development of metastases from distant cancers. This support is mediated through cell–cell contact, release of soluble mediators and formation of extracellular matrix. By using a proteomic approach, we characterized the protein release by in vitro cultured human MSCs (10 donors) and osteoblasts (nine donors). We identified 1379 molecules released by these cells, including 340 proteins belonging to the GO-term Extracellular matrix. Both cell types released a wide range of functionally heterogeneous proteins including extracellular matrix molecules (especially collagens), several enzymes and especially proteases, cytokines and soluble adhesion molecules, but also several intracellular molecules including chaperones, cytoplasmic mediators, histones and non-histone nuclear molecules. The levels of most proteins did not differ between MSCs and osteoblasts, but 82 proteins were more abundant for MSC (especially extracellular matrix proteins and proteases) and 36 proteins more abundant for osteoblasts. Finally, a large number of exosomal proteins were identified. To conclude, MSCs and osteoblasts show extracellular release of a wide range of functionally diverse proteins, including several extracellular matrix molecules known to support cancer progression (e.g., metastases from distant tumors, increased relapse risk for hematological malignancies), and the large number of identified exosomal proteins suggests that exocytosis is an important mechanism of protein release.

## 1. Introduction

The cellular elements of the bone marrow microenvironment consist of hematopoietic cells together with various stromal cells that support normal as well as leukemic hematopoiesis and metastatic disease [1,2]. The hematopoietic cells have a hierarchical organization, and minor populations of immature stem cells are essential to maintain both normal and leukemic hematopoiesis [3,4]. These stem cells are localized to stem cell niches, i.e., specialized bone marrow microenvironments that are important to maintain and regulate the function and homeostasis of stem cells [5,6]. These niches have traditionally been divided into two main types, the osteoblastic or endosteal and the vascular or sinusoidal niche. These two microenvironments seem to have different functions with regard to regulation of stem cell homeostasis, i.e., quiescence and maintenance versus proliferation/differentiation, respectively [3]. Recent studies have also identified an arteriolar niche where periarteriolar stromal cells are important. Thus, osteoblasts as well as mesenchymal stem/stromal cells (MSCs), together with other nonleukemic cells [6], are regarded as members of the stem cell niches, but their importance will possibly differ between various bone marrow compartments [3]. These stromal cells may thereby become important for the migration and localization of stem/progenitor cells to various bone marrow microcompartments.

The extracellular matrix is also important for regulation of hematopoiesis. The matrix proteins are important both for communication between bone marrow cells (e.g., as a reservoir of growth factors) and for regulation of hematopoietic cell migration and localization [1,3]. The stromal cells within the bone marrow are therefore important for the microenvironment both directly through their release of various mediators/proteins and indirectly through the formation of extracellular matrix [7]. Thus, the stromal cell support of normal and leukemic hematopoiesis, and possibly also cancer metastases, seems at least partly to be mediated through their contribution to the extracellular matrix, including the matrix in stem cell niches.

Several studies suggest that MSCs and osteoblasts are important in carcinogenesis and therefore also for chemosensitivity and/or cancer progression. First, as discussed in a previous review [8] MSCs seem to contribute to development of bone marrow metastases at least in certain experimental models; the MSCs then contribute to a microenvironment that stimulates homing of circulating tumor cells to the bone marrow and thereafter promotes the development of metastases both through adhesion-dependent and adhesion-independent (i.e., mediated by soluble mediator release) and adhesion-dependent mechanisms [8]. Osteoblasts seem to facilitate development of metastases through the same mechanisms, and this progression may occur after an initial period of cancer cell dormancy [9,10,11]. Furthermore, both MSCs and osteoblasts are able to support the development of hematological malignancies and confer chemoresistance; this seems to be true both for myeloid and lymphoid malignancies [8,12,13]. Thus, these stromal cells seem to be important both for the progression of distant tumors to metastatic bone marrow disease [8,9,10,11,12,13] and the progression of hematological malignancies to from initial remission induction to later relapse [14]. In this context, we compared the extracellular protein release by bone marrow derived MSCs and osteoblasts. Even though their molecular release profiles show many similarities, the two cell types differ especially with regard to their release of certain extracellular mediators that are important regulators of hematopoiesis.

## 2. Results

### 2.1. Osteoblasts and Bone Marrow MSCs Release a Wide Range of Diverse Proteins

Osteoblasts were derived from nine and MSC from 10 healthy individuals; the characteristics of these donors are presented in Table 1. These enriched cells were cultured in serum-free medium for 48 h before supernatants (i.e., conditioned media) were analyzed. A total of 1379 proteins reached detectable levels either for MSC or osteoblasts.

We were able to quantify 1255 proteins for the various MSCs. The number varied between 470 and 1004 proteins for individual cell donors; 795 proteins (63%) were quantified for at least five MSC donors and more than 400 proteins were quantified for at least nine of the 10 donors (Figure 1, upper left). The biological reproducibility was good, with Pearson R correlation generally being >0.85 (ranging from 0.92 for MSC1-CM vs. MSC3-CM to 0.33 for MSC6-CM vs. MSC10-CM; see Figure 1, lower left). However, donor MSC6 correlated poorly with the other donors.

For the osteoblast samples, we quantified 1217 proteins; the number also varied between donors (range 491–939); 571 (46.8%) proteins could be quantified for at least five donors (Figure 1, upper right). The biological reproducibility was generally good also for these samples (Pearson R > 0.70) except for donor OB4 (Figure 1, lower right). For the quality control samples 724 (61.7%) proteins were quantified in all samples and the release of these proteins was highly correlated.

We finally did a hierarchical clustering analysis of the Pearson correlation coefficients for the MSC and osteoblast donors (Figure 2). This analysis showed that MSCs and osteoblasts clustered separately except for the two outliers MSC6 and OB4 that differed from the other samples.

### 2.2. A Minority of MSCs and Osteoblasts Showed Abnormal Release Profiles

Relatively few MSC6 cells attached to the culture flask after they had been seeded, and these cells had to be expanded for 7 days before they reached 70% confluence, and the culture medium could be replaced with serum-free medium. For the majority of MSCs, the medium could be changed after 4–5 days. Only 5–10% of the MSC6 cells were alive (i.e., bright, adherent cells) after 48 h of culture in serum-free medium (i.e., at the time when supernatants were harvested), and the viable cells were relatively small compared with cells of the other MSC donors. Based on light microscopy the large majority of cells were viable and adherent both during at the end of culture for all the other MSC donors. Cells from MSC4 and MSC9 had to be cultured for an extended time period of 13 days before reaching 70% confluence, but in contrast to MSC6, cells from these two MSC donors had similar morphology (i.e., viable and adherent) as the other MSCs both at the time of medium change and at the time of supernatant harvesting (Figure 1 and Figure 2). Thus, MSCs derived from various healthy individuals differ in their in vitro growth characteristics, but differences in the protein release profile seem to be less common and were only observed for the outlier MSC6 donor.

The OB4 donor differed from the other osteoblast donors with regard to the protein release, but these cells did not differ with regard to morphology or cell proliferation during culture (Figure 1 and Figure 2).

### 2.3. Both MSCs and Osteoblasts Show Extracellular Release of a Large Number of Functionally Diverse Proteins

We identified a total of 1379 proteins for the MSCs and the osteoblasts. The released proteins were very heterogeneous both for the MSCs and the osteoblasts, and more than 90% of them did not differ significantly between the two cell types when comparing the overall results (see Section 2.5). A detailed list of important protein main classes/subclasses including at least five proteins is given in Appendix A, and a summary of this list is presented in Table 2. First, extracellular matrix molecules constituted a large group of released molecules; the majority of these proteins were fibrillar and nonfibrillar collagens although several other extracellular matrix molecules were also released. A complete list of proteins released by MSCs and/or osteoblasts belonging to the GO term Extracellular Matrix is given in Appendix A. Second, a large number of enzymes were released, including several proteases (also complement factors) but also protease inhibitors. Several of the released proteins are also important for protein stabilization and posttranscriptional modification. Third, several proteins with important intracellular functions were released including (i) proteins involved in intracellular transport and/or exocytosis (e.g., cytoskeleton, 14-3-3 proteins and V-ATPase proteins), and (ii) molecules involved in protein synthesis, together with (iii) nuclear proteins (e.g., histone and non-histone chromatin), whereas (iv) proteins involved in regulation of cellular metabolism constituted a relatively small part of the released proteins. Several of the released molecules reflect the function of GTP. Finally, relatively few cytokines, soluble cytokine receptors and soluble adhesion molecules were detected, although our observations suggest that IGF (release of IGFALS and IGFBP7) may be of particular interest. Taken together, these results show that a large number of structurally and functionally diverse proteins originating from various intracellular compartments are constitutively released both by MSCs and by osteoblasts.

Many previous studies of the human MSC protein profiles have investigated relatively few cell donors and used alternative proteomic methods and thereby detecting fewer proteins [15]. Three previous studies have investigated the MSC secretoma by using a methodological strategy allowing the identification of a large number of proteins [16,17,18]. However, all these studies investigated few cell donors and/or only young MSC donors. The overall proteomic profiles in these three studies showed similarities with our present results; a large number of released proteins had a known extracellular localization (including collagens and laminins), intracellular cytoplasmatic proteins and nuclear proteins (histones as well as other nuclear protein). Thus, a similar diversity of the MSC secretome profile with regard to cell compartment has also been described in previous studies, and in our present study we in addition observed a similar diversity for the osteoblastic secretome.

### 2.4. Only a Heterogeneus Subset of MSC- and Osteoblast-Released Proteins Reach Detectable Levels for most MSC or Osteoblast Donors

The results described above clearly show that both MSC and osteoblasts derived from different donors vary with regard to the number of detectable proteins, and both cell types are able to release a high number of structurally as well as functionally diverse proteins. However, the data presented in Figure 1 (upper part) shows that there is a subset of proteins that reach detectable levels for all or most donors. The data presented in Appendix A show that 426 heterogeneous proteins (34% of all MSC-released proteins) could be detected for at least nine of the 10 MSC donors. Similarly, 317 proteins were released by at least eight of the nine osteoblast cell populations (26% of all osteoblast-released proteins), and these proteins also showed a considerable diversity (Appendix A). Finally, 269 structurally and functionally heterogeneous proteins (20% of all the 1379 identified proteins) were detected both for at least nine MSC and eight osteoblast cell populations (Appendix A).

We investigated the release of selected protease and protease inhibitors (Serpin-C1, Serpin-E1, Cystatin-B, Cystatin-C, CD147, CCFD, Neutrophil elastase, MMP1, MMP2) for three MSC donors; these cells were purchased from another distributor and were cultured in an optimal serum-containing medium [19]. When using antibody-based analyses [19], the serpins and cystatins showed relatively high levels (usually >1000 pg/mL). High levels were also seen for Serpin-E1, Cystatin-B, Cystatin-C, MMP1 and MMP2 in our present study, whereas we could not detect Serpin-C1. The three last mediators showed low (i.e., <1000 pg/mL) or undetectable levels when tested in this alternative experimental model and also in our present proteomic study. Furthermore, the MSC10 cells were investigated in an independent experiment and showed a similar protease/protease regulator profile and overall protein release profile in both experiments.

### 2.5. A Diverse Minority of Constitutively Released Proteins Differ between MSCs and Osteoblasts

We identified 82 proteins that were released at significantly increased levels by MSCs, i.e., either (i) being detected for at least three MSCs and three osteoblast donors and the corresponding statistical comparison showing *p* < 0.05 and at the same time a significant fold change (FC) (*p* < 0.05) determined by Z-score statistics; or (ii) being detected either in one or none of the osteoblast but at least seven of the MSC populations. These proteins are listed in Appendix A, they are described in Appendix A, and they are classified in Table 3. It can be seen that these proteins are involved in the regulation of a wide range of fundamental cellular functions, especially communication between neighboring cells and interactions between cells and extracellular stroma. First, several extracellular matrix proteins showed increased release by MSCs (Table 3, Appendix A), including 11 collagens and nine other extracellular matrix molecules. Several proteases also differed, and some of these are involved in the modulation/cleavage of extracellular matrix molecules. These matrix molecules are important both for cell–cell and cell–matrix interactions. The cadherins seem to be of particular interest as two cadherins and two cadherin-interacting molecules differed significantly. Second, several of the significantly increased proteins are important for protein stabilization and modulation, including chaperones together with enzymes involved in posttranscriptional modulation or proteolytic cleavage. Third, with regard to communication between neighboring cells. TGFβ as well as four proteins involved in TGF-initiated signaling were significantly increased together with four proteins involved in angiogenesis. Finally, ten released proteins are involved in cellular metabolism, especially proteins important for lipid/cholesterol metabolism and mitochondrial functions.

Osteoblasts showed increased release of only 36 proteins when using similar criteria as described for MSCs (i.e., (i) *p* < 0.05 plus FC significance <0.05, or (ii) expressed by ≤1 MSC but at least eight osteoblast populations; see Appendix A). Twenty-two of the 36 significant proteins were annotated to the extracellular space, and two to the insulin-like growth factor ternary complex (IGFBP3 and IGFALS). Half of the proteins were annotated as Secreted, which was comparable to the 43% found among the upregulated MSC proteins.

A total of 118 (82 and 36, respectively) proteins showed abundant expression when comparing MSCs and osteoblasts; both the total number of abundantly expressed (118, out of 1379, Fischer’s exact test, *p* = 0.0007) and the unequal distribution of abundantly expressed proteins between the two cell types (*p* = 0.0011) illustrate that these proteins were not identified by coincidence alone.

We did a GO term analysis of the proteins mainly released by MSCs. A total of 746 proteins could be quantified for at least three donors/samples of each cell type (i.e., 746 proteins), and the GO term analysis was based on those among these 746 proteins that differed significantly between MSCs and osteoblasts (Welch’s *t*-test) plus those proteins that were detected only or mainly for MSC (i.e., ≤1 osteoblast donor/sample, ≥7 MSC donors/samples; see above). The most significant GO terms identified for proteins increased in MSC reflected mainly differences with regard to extracellular matrix, cell adhesion, cell migration (Appendix A). The proteins increased in osteoblasts based on similar criteria did not significantly reflect any GO terms (data not shown).

We finally did a network analysis based on both the MSC and osteoblast proteins (Figure 3); the identified networks identified the same differences as the GO term analyses (e.g., extracellular matrix), but in addition this analysis illustrates the diversity of the identified proteins.

### 2.6. Most of the Released Proteins Show only Quantitative Differences between MSC and Osteoblasts

Only a minority of the significantly differing proteins showed a qualitative difference between MSCs and osteoblasts, i.e., could be detected for a majority of one cell type but not for the other cell type (Table 4). Sixteen proteins were only detected for MSCs, whereas six proteins were detected only for the osteoblasts. Among the MSC proteins were the collagen alpha 1 (X) chain, lysyl oxidase homolog 3, and the HGF growth factor, whereas the collagen alpha 1 (II) chain was released at detectable levels only by osteoblasts.

### 2.7. Both MSCs and Osteoblasts Release a Wide Range of Molecules that Are Involved in the Regulation of Normal Hematopoiesis

Several molecules expressed by normal bone marrow stromal cells are important for regulation of normal hematopoiesis, including extracellular matrix molecules, proteases, adhesion molecules released in biologically active soluble forms, other cell surface molecules and cytokines/cytokine regulators; the stromal cells may in addition be involved in the transport and homeostasis of metal ions that are involved in the regulation of hematopoiesis [1,2,3,5]. These molecules are also important for the formation of the various stem cell niches. Previously, characterized molecular regulators of normal hematopoiesis that could be released by in vitro cultured MSCs and osteoblasts are listed in Table 5. It can be seen that both cell types release a wide range of molecules that are involved in regulation of normal hematopoiesis. Most of these mediators did not differ when comparing the two cell types, with the exceptions being the seven mediators—MMP11, LOX, one of the cadherins, proteoglycan, hyaluronic acid, TGFBP and UPA—that showed higher expression of MSCs.

Cluster 5 in the protein network included 16 proteins annotated to platelet degranulation (Figure 3, Appendix A). Most of these proteins are expressed in several different organs/tissues and thus reflect basic cellular functions rather than a specific function in platelets (Appendix A). These 16 proteins included several metabolic regulators together with protease inhibitors, cell surface molecules and growth regulators.

### 2.8. Variation in Protein Release between Individual MSC and Osteoblast Donors

To further investigate the heterogeneity of MSCs and osteoblast donors we performed an unsupervised hierarchical cluster analysis based on the stem cell regulatory proteins listed in Table 5 (Figure 4) [1,2,3,4,5]. It can be seen that the expression of these proteins varied between patients. We could identify two main clusters including most MSC and osteoblast donors, respectively. Even though most MSC and osteoblast donors were included in each of these two main clusters, it can be seen that there is a variation between donors. Some of the donors localized separately as outliers (the two MSC5/MSC6 donors, one osteoblast donor), and we could also identify subsets/subclusters both for the main MSC and osteoblast cluster. This analysis shows that he heterogeneity in protein profiles between individual donors involves their stem cell (niche) supporting functions.

We also compared male and female donors with regard to their proteomic release profiles. These observations should be interpreted with great care because the analyses are based on relatively few donors in each of the compared groups (only four female donors). However, it is a relevant and possibly important question because differences between male and female MSCs have been described previously and may be important especially with regard to MSC aging [20]. For MSCs only 22 proteins showed significant differences between male and female donors, but all except one of these proteins reached only borderline significance (0.01 < *p* < 0.05) and no protein network could be identified (data not shown). For osteoblasts 48 proteins differed significantly, and we could also identify several protein interaction networks (Appendix A).

### 2.9. Mechanisms of Extracellular Protein Release in MSCs and Osteoblasts; Exosomes vs. Proteolysis

Exosome release is one of the mechanisms for communication between cells, and 683 of the identified proteins belonged to the GO term exosome (Appendix A). Seventy-three of these proteins belong to the top 100 exosomal proteins (http:/www.exocarta.org/exosome_marker-new), and 21 of them are among the top 25 proteins. None of these proteins showed any difference between MSCs and osteoblasts. Although we did not isolate and characterize exosomes in our present study, this large protein number strongly suggests that extracellular protein release through exosomes is important for the communication between MSCs/osteoblasts and their neighboring cells.

A total of 206 proteins included in the term Protease reached detectable levels in our study (Appendix A), and most of these proteases could be detected both for MSCs and osteoblasts. These enzymes may be important for extracellular modulation of other released molecules and for the release of cell surface molecules (Appendix A), but despite this protease release relatively few soluble adhesion molecules and cytokine receptors were detected in our MSC/osteoblast supernatants. These last observations suggest that proteolytic cleavage of cell surface molecules also contributes to protein release of both these cell types, although exosomal release seems to be more important.

## 3. Discussion

Nonleukemic stromal cells support both normal and leukemic hematopoiesis, and probably also cancer metastases, through their release of soluble mediators and extracellular matrix molecules as well as through cell–cell contact [3]. Both normal and leukemic hematopoietic cell populations have a hierarchical organization, and the stem cells with self-renewal capacity are supported by the non-hematopoietic cells in the stem cell niches [5]. These niches include osteoblasts and MSCs together with endothelial cells and monocytes [3,6]. In this study, we compared the extracellular release of various proteins to this common microenvironment by osteoblasts and MSCs. 

We investigated enriched MSCs and osteoblasts derived from bone marrow, and we observed differences between individuals with regard to proliferation (MSCs), survival (MSCs) and protein release profile (two MSC and one osteoblast donor; see Figure 2 and Figure 4). However, all our donors were randomly selected from the manufacturer’s list of available donors, and we could not detect any differences between the samples with regard to initial handling, cellular characteristics (including functional studies) and thawing/culture. For these reasons, we decided not to leave out any of the donors showing exceptional characteristics during culture because this may reflect a true biological heterogeneity between donors. One possible explanation for such heterogeneity could be different distributions of MSC subsets [20,21,22,23]. Another possible explanation at least for MSCs and therefore also for the MSC derived osteoblasts could be differences in MSC aging between our donors [20,21,22,23]. The exceptional MSC6 donor was one of the oldest donors, and previous studies have shown that MSC aging is associated with decreased proliferative capacity, induction of senescence, and altered expression of cell surface molecules with modulation of cellular migration [20,21,22,23].

MSCs and osteoblasts represent two cell subsets among the bone marrow stromal cells; even though they represent minor subsets of bone marrow cells their importance in normal as well as leukemic hematopoiesis is based on their localization in bone marrow stem cell niches with MSCs mainly localized in a perivascular niche whereas osteoblasts are localized to the endosteal niche [24]. Their protein release profiles showed several similarities; most of the detected proteins could be released by both cell types and only a minority of the proteins were released at detectable levels only or mainly by one of them (Table 4). The explanation for this is probably that osteoblasts can differentiate from MSCs; this differentiation can be influence by several factors including activation of the cellular energy sensor AMPK (AMP-activated protein kinase), phosphoinositide-3-kinases (PI3K), mammalian target of rapamycin C2 (mTORC2) and Jak-2 protein tyrosine kinase or inhibition of STAT3 inhibition [25,26,27,28,29]. Thus, despite their differences in bone marrow localization and stem cell support by MSCs and osteoblasts, the protein release profiles of the two cell types show many similarities.

We investigated MSCs and osteoblasts in an experimental in vitro model, and we would emphasize that MSCs and probably also osteoblasts can be influences by several additional factors in cancer patients. The hypoxia of the bone marrow microenvironment can influence both autophagy, survival, proliferation and migration of MSCs [30,31]. Additional factors can also be important in cancer patients; cancer-associated modulation of bone marrow MSCs can modulate the chemosensitivity of malignant cells and thereby influence the progression to relapse [14], and exposure to various cytotoxic drugs may also affect the functional status of MSCs [32]. For these reasons, our present results have to be interpreted with care.

We compared the levels of selected proteases/protease regulators detected by proteomic analyses and by antibody-based analyses, and similar variations were then observed except for Serpin-C1. However, the culture conditions differed for these two analyses, and the more optimal serum-containing medium used in the ELISA studies may explain the divergent observations for the single mediator Serpin-C1. Furthermore, for MSC10 a similar proteomic profile variation was observed in two independent analyses. Finally, our proteomic profiles showed a similar diversity as described before in smaller studies of younger individuals. Our present proteomic results are thus not in conflict with available results from previous MSC studies, but our present studies include a larger number of cell donors, a detailed comparison between MSCs and osteoblasts and a focus on individual variation and cancer cell support.

Cells can release proteins through various pathways. First, proteins can be released through secretion or exocytosis, including the release of free soluble molecules but also exosomal release that has been described in previous MSC studies [33,34,35,36,37,38]. In our present study, we cannot distinguish between these two mechanisms; due to our strategy for sample preparation, we identified proteins both released in exosomes and soluble proteins in the culture medium. Second, soluble forms of cell surface molecules can be released after proteolysis [39,40,41]. We identified a wide range of proteases in the culture supernatants both for MSCs and osteoblasts, and this may therefore be important for the release of biologically active and soluble cell surface molecules. However, this is not the only mechanism for the release of cell surface molecules, because truncated forms may also be synthesized intracellularly and released through secretion [42]. Finally, intracellular molecules may be released by apoptotic/necrotic cells. In our opinion, the impact of this last mechanism is less important because most of our cell populations were proliferating and showed no morphological signs of cell death during in vitro culture.

We cannot exclude the possibility that some of the proteins showing significant differences between MSC and osteoblasts have been identified by coincidence, but coincidence cannot explain all our observations, and in our opinion, proteins identified by coincidence must be a minority of the abundant proteins. First, both the total number of differentially expressed protein (118 out of 1379) and the larger number of increased expression in MSC compared with osteoblasts (82 vs. 36) is significantly different from what would be expected by coincidence. Furthermore, both GO term and network analyses as well as the detailed descriptions of individual proteins show that the abundant proteins are highly selected and do not include or include very few proteins from large protein classes/subclasses, e.g., exosomal proteins, histones and non-histone nuclear proteins.

Both MSCs [17,19,43] and osteoblasts [12,44,45,46,47,48] release a wide range of soluble mediators, including interleukins, chemokines and various growth factors. However, relatively few cytokines (only TGFβ and HGF) could be detected by our proteomic approach, and this is probably due to the lower sensitivity of our proteomic assay compared with various antibody-based assays. Furthermore, our present study confirmed a link between the coagulation system and modulation of the extracellular matrix [1], especially through the constitutive release a plasminogen activator. Our present study also suggests that locally released complement factors may be involved in the modulation of extracellular proteins because these factors were released both by MSCs and osteoblasts. Previous studies have also described local release of complement factors by mesenchymal cells [49,50,51,52].

Several extracellular matrix molecules and other molecules expressed on or released from the cell surface of bone marrow stromal cells are important for the regulation of normal hematopoiesis; this includes a wide range of various extracellular matrix molecules, adhesion molecules and other cell surface molecules released in biologically active soluble forms as well as various soluble mediators [1,2,3,5]. Several enzymes can also be released and then influence hematopoiesis through cytokine activation, cleavage of matrix molecules, or modulation of extracellular matrix (e.g., lysyl-oxidase mediated molecular crosslinking) [53,54,55]. Several intracellular mediators are also released, but their possible extracellular functions have not been characterized. Most of these molecules were released at similar levels by MSCs and osteoblasts, although a minority showed increased release by MSCs. Such quantitative differences in the release of extracellular regulators may contribute to the previously described differences among various stem cell niches in their regulation of normal hematopoiesis [3]. Many of the extracellular regulators involved in normal hematopoiesis are also important in leukemic hematopoiesis although their function in leukemic hematopoiesis is less well characterized; e.g., collagen/integrin/Rac1 [56,57], the CDH1 [58] and CDH2 cadherins [59], thrombospondin [60,61], TGFβ [62,63,64], osteopontin [65] and LOX/lipid metabolism [66,67] all seem to be important for leukemogenesis and/or chemosensitivity in hematological malignancies.

Studies of several malignancies suggest that the extracellular matrix is important for the development and chemosensitivity of human malignant cells. Firstly, an extracellular matrix profile identified a subset of breast cancer patients with increased risk of progression [68,69]. This profile included high levels of SPARC, COL1A1, COL5A2, COL6A3, LAMA4 and MMP11; all these molecules were detected both for MSC and osteoblasts, but COL1A1 and COL5A2 were significantly increased for MSCs. Furthermore, an extracellular matrix gene cluster including SPARC, COL1A1, COL1A2, FN1, LOX and TIMP3 was associated with tamoxifen resistance in patients with metastatic breast cancer [70]; all of these except TIMP3 were detected both for MSCs and osteoblasts in our proteomic study with COL1A1 and FN1 being significantly higher for MSCs. Second, signatures increased in collagen expression together with LOX, THBS2, TIMP3 and SPARC have also been associated with decreased survival in ovarian cancer [71,72]. Third, the metastatic potential in colon cancer is associated with expression of extracellular matrix-encoding genes [73]. Finally, certain lymphoproliferative diseases are also associated with expression of COL6A1, COL18A1, MMP3 and TIMP1 [74], and a subset of patients with diffuse large B cell lymphoma and adverse prognosis show decreased expression of POSTN, SPARC, COL1A1, COL3A1,CSTK, MMP9 and LAMB3 [75]. Thus, disease progression and chemosensitivity of solid tumor metastases as well as hematological malignancies (i.e., liquid tumors) are associated with specific extracellular matrix profiles that may contribute to the aggressive/resistant phenotype, but the effect of individual genes may differ among diseases.

A wide range of proteins involved in important intracellular processes were also released both from MSCs and osteoblasts. Histones provide structural stability to the chromatin and are transcriptional regulators, but histones may also be released into the extracellular space either freely, as DNA-bound nucleosomes or as a part of extracellular traps [76]. These extracellular forms serve as damage-associated molecular pattern proteins through their interactions with Toll-like receptors that can be expressed by several normal cell types [76], but also by malignant cells [77]. Other nuclear proteins can also be released and are even regarded as possible therapeutic targets in certain human diseases [78]. Furthermore, heat shock proteins seem to have additional important extracellular functions; they can be involved in cell–cell communication and seem to be released by non-classical pathways, including exosomes [79]. Other secreted chaperones seem to be important through their binding to extracellular client proteins and may even contribute to their degradation through increased endocytosis [79]. The possible extracellular functions of many of the other released intracellular molecules are not known, but our present observations suggest that these molecules should be further investigated for their possible roles in exosome-mediated communication between stromal cells and normal hematopoietic, leukemic or metastatic cancer cells. The intracellular proteins are probably not released simply due to cell death because one would then expect a random distribution and not an absence of for example mitochondrial proteins.

As discussed above, cells can release proteins to the extracellular space through various mechanisms, including proteolysis of cell surface proteins and release through exosomes [80]. We cannot evaluate the relative contribution of each mechanism in our present study because we did not isolate and analyze exosomes separately. However, a very large part of our detectable proteins are included in the GO term exosomes, thereby suggesting that exosomal release is an important mechanism. However, the extracellular release of a large number of proteases suggests that the alternative mechanism of proteolytic cleavage of surface molecules is also possible [39,40,41,81,82,83,84,85,86,87], even though very few soluble adhesion molecules or cytokine receptors were detected in our present study. Furthermore, exosomes can influence their target cells either through interactions with signaling receptors, fusion with the cell membrane followed by delivery of their content to the cytosol, or the exosomes become internalized and merge with endosomes to either undergo transcytosis or maturation into lysosomes [88]. Thus, the intracellular delivery of exosomal molecules may then influence specific cellular processes as suggested by the protein networks illustrated in Figure 3.

Even though our MSC and osteoblast donors showed a limited heterogeneity with regard to the overall protein release profile, some differences were observed, and these differences also included the release of proteins important for the function of the stem cell niches/normal hematopoietic stem cells (Figure 1, Figure 2 and Figure 4). Previous studies have also described differences in MSC between male and female donors especially in elderly individuals [20]. Our present cell donors should be regarded as elderly [20], and our present results suggest that differences between men and women can also be detected for the protein release profiles of osteoblasts. 

In the present study we have investigated bone marrow MSCs and osteoblasts derived from healthy individuals. An important question is therefore how the secretoma of MSCs and osteoblasts is modified by the crosstalk between these stromal cells and the malignant in their common bone marrow microenvironment. The communication between these cells is probably mediated both by direct cell–cell contact, modulation of the extracellular matrix and the release of soluble mediators [41,89]. As illustrated by our present results, all three mechanisms can be influenced by the stromal cell secretoma, e.g., through the release of biologically active adhesion molecules, altered release or modulation of matrix molecules, altered release of soluble mediators or the release of their soluble receptors. Furthermore, one also has to address the question of patient heterogeneity; this is illustrated by our present studies of acute myeloid leukemia (AML) showing that patients are very heterogeneous with regard to the secretoma-mediated communication between leukemic and stromal cells [13,19,43,90,91]. These previous observations are also supported by our own preliminary proteomic studies of the crosstalk between normal MSCs and primary AML cells (Aasebø, unpublished data). Our present study should therefore be regarded as a basis for future studies of secretoma-mediated crosstalk between stromal and cancer cells in the bone marrow microenvironment and the effect of this crosstalk on stromal cell protein release.

Conditioned medium from MSCs are now considered as a possible therapeutic tool [34,36,92], and the same is true for MSC derived exosomes [93]. The conditioned medium probably includes exosomes and a wide range of various soluble mediators, e.g., proteins, metabolites or nucleic acids. Our present study represents a broad characterization of proteins released by MSCs and osteoblasts during in vitro culture and contributes to the scientific basis for the further development of this potential therapeutic strategy. Other studies suggest that detection and molecular analysis of exosomes could be used as biomarkers in cancer patients [94]. However, due to our methodological strategy we would emphasize that we have analyzed all proteins released by the cells, including proteins released in exosomes. The route of release (exosomes vs. exocytosis) will be important for the mechanisms of action and the final biological effects of the proteins as described above, and the protein effects will also be influenced by the other constituents of the exosomes, e.g., lipids as well as other metabolites together with mRNA [80,81,88,93].

The effects of soluble mediators derived from MSC and osteoblasts have been studied more in detail in AML. By using a coculture model where stromal and AML cells were separated by a semipermeable membrane it was possible to study the effect of the crosstalk between the two cell types mediated by soluble mediators. These studies showed that the stromal cell secretoma could increase proliferation and/or promote survival of the malignant cells; this was true both for normal MSCs, osteoblasts and fibroblasts but the mechanisms behind this effect differed between patients [12,13,95]. However, protein communication, including communication through the release of exosomes, is not the only interaction between stromal and leukemic cells; communication through miRNA release [96] as well as metabolic modulation [97] also seems to contribute to the non-adherent crosstalk between normal stromal and leukemic cells. These observations illustrate the complexity of cellular communication, and a similar complexity may also be present in other malignancies. The possible modulation of the MSC/osteoblast secretoma by bidirectional crosstalk between stromal and cancer cells further adds to this complexity [98]. However, despite this complexity it should be emphasized that the final effects in functional experimental studies is AML cell support with increased proliferation/survival.

To conclude, bone marrow MSCs and osteoblasts release a wide range of functionally divergent proteins that are involved in the communication between cells, the regulation of normal as well as leukemic hematopoiesis and possibly also development of metastases from distant tumors. However, many proteins involved in the intracellular regulation of fundamental cellular functions are also released, and the possible extracellular function of these proteins should be further investigated.

## 4. Material and Methods

### 4.1. Human MSCs and Osteoblasts

Cryopreserved human bone marrow MSCs (at least 500,000 cells) derived from 10 healthy donors and human osteoblasts (at least 500,000 cells) derived from nine healthy donors were purchased from PromoCell GMBH (Heidelberg, Germany; reference codes C-12974 and 12720, respectively) (Table 1). The cells had been cryopreserved in passage two after being characterized by flow cytometric analysis and in addition examined for morphology, proliferative capacity, adherence rate and viability. All samples showed a similar high viability and proliferative capacity before freezing, and they were cryopreserved according to the same highly standardized procedure. The MSCs were in addition tested for their ability to differentiate into mesenchymal lineages, and the osteoblasts were tested for expression of alkaline phosphatase and capacity of mineralization (manufacturer’s information). These functional characteristics were confirmed for all the MSC and osteoblast samples. Mycoplasma infections could not be detected for any sample. The samples were stored in liquid nitrogen and later handled according to the manufacturer’s instructions.

### 4.2. Cell Culture

The study was approved by the Regional Ethics Committee (REK Vest 350/2017). MSCs and osteoblasts (1 mL) were thawed at 37 °C according to a highly standardized procedure recommended by the manufacturer, and approximately 1.5 × 10^5^ cells were further expanded in T25 flasks (Falcon; Glendale, AZ, USA) with Mesenchymal Stem Cell Growth Medium (Sigma-Aldrich, Saint- Louis, MO, USA; supplied by PromoCell ref. no. C-28009) and Osteoblast growth medium (Sigma-Aldrich; supplied by PromoCell ref. no. C-39615) with supplement-39615 (Sigma-Aldrich; supplied by PromoCell ref. no. C-27001), respectively. The culture medium was exchanged after 24 h. When the cells had reached 70% confluence, the culture medium was exchanged with serum-free IMDM medium (ThermoFisher Scientific; Waltham, MA, USA). The cultures were thereafter incubated for 48 h before supernatants were collected and centrifuged at 3000× *g* for 10 min before being aliquoted and stored at −80 °C. These conditioned media samples were later concentrated by ultra-filtration using Amicon Ultra-15 3kDa cutoff centrifugal filters (Millipore; Burlington, MA, USA) prior to proteomic sample preparation. The filters were conditioned with 4 mL H_2_O (HPLC grade) before adding the conditioned medium, which was centrifuged for 30–45 min at 3220× *g* at 4 °C. dH_2_O (4 mL) was then added for washing, using the same settings and repeated twice. Protein concentration was measured using the Qubit Protein Assay (Thermo Fisher Scientific; Waltham, MA, USA). 

### 4.3. Proteomics Sample Preparation

The conditioned media samples containing 10 μg of protein were buffered with 50 mM Tris/1 mM CaCl, reduced with 20 mM DTT (7 min at 95 °C), alkylated with 50 mM iodoacetamide (1 h at RT, (dark), and digested overnight at 37 °C with 1:50 enzyme: substrate ratio of sequencing grade trypsin (Promega, Madison, WI, USA). Following digestion, samples were acidified with methanoic acid and desalted using HLB Oasis SPE cartridges (Waters; Milford, MA, USA). Samples were eluted with 80% acetonitrile in 0.1% methanoic acid and concentrated (speedvac). The peptides were stored at −80 °C until use. A quality control sample was also made by pooling conditioned media from the MSC and osteoblast samples, in addition to conditioned media from in vitro cultured primary acute myeloid leukemia (AML) cells. This quality control sample was analyzed after every tenth sample in the following LC-MS analysis to document reproducible instrument performance.

The final cultures were prepared in IMDM medium alone with supplementation of serum, phenol red, lipids, growth factors or proteins. We could not detect albumin or immunoglobulin (i.e., markers of serum contamination) in our proteomic analyses of conditioned media; these proteins are detected at high levels in normal serum but could not be detected in our culture supernatants.

All cultures were regularly evaluated by light microscopy during and at the end of the culture period; the large majority of cells were viable (i.e., bright and adherent) for all donors during culture but not for MSC6 at the end of culture.

### 4.4. Liquid Chromatography (LC) Tandem Mass Spectrometry (MS) Analysis

About 0.5 μg protein as tryptic peptides dissolved in 2% acetonitrile (ACN), 0.5% formic acid (FA), were injected into an Ultimate 3000 RSLC system (Thermo Scientific, Sunnyvale, CA, USA) connected online to a linear quadrupole ion trap-orbitrap (LTQ-Orbitrap Elite) mass spectrometer (Thermo Scientific, Bremen, Germany) equipped with a nanospray Flex ion source (Thermo Scientific, Sunnyvale, CA, USA). The sample was loaded and desalted on a pre-column (Acclaim PepMap 100, 2 cm × 75 µm ID nanoViper column, packed with 3µm C18 beads) at a flow rate of 6 µL/min for 5 min with 0.1% TFA (trifluoroacetic acid, vol/vol).

Peptides were separated during a biphasic ACN gradient from two nanoflow UPLC pumps (flow rate of 200 nL/min) on a 50 cm analytical column (Acclaim PepMap 100, 50 cm × 75 µm ID nanoViper column, packed with 2 µm C18 beads). Solvent A and B were 0.1% FA (vol/vol) in dH_2_O and 100% ACN respectively. The gradient composition was 5% B during trapping (five minutes) followed by 5–7% B over one minute, 7–32% B for the next 129 min, 32–40% B over 10 min, and 40–80% B over 5 min. Elution of very hydrophobic peptides and conditioning of the column were performed during 20 min isocratic elution with 80% B and 20 min isocratic elution with 5% B respectively. The eluting peptides from the LC-column were ionized in the electrospray and analyzed by the LTQ-Orbitrap Elite. The mass spectrometer was operated in the DDA-mode (data-dependent-acquisition) to automatically switch between full scan MS and MS/MS acquisition. Instrument control was through Tune 2.7.0 and Xcalibur 2.2.

Survey full scan MS spectra (from m/z 300 to 2000) were acquired in the Orbitrap with resolution R = 240,000 at m/z 400 after accumulation to a target value of 10^6^ in the linear ion trap with maximum allowed ion accumulation time of 300 ms. The 12 most intense eluting peptides above an ion threshold value of 3000 counts and charge states 2 or higher were sequentially isolated to a target value of 10^4^ and fragmented in the high-pressure linear ion trap by low-energy CID (collision-induced dissociation) with normalized collision energy of 35% and wideband-activation enabled. The maximum allowed accumulation time for CID was 150 ms, the isolation was maintained at 2 Da, activation *q* = 0.25, and activation time of 10 ms. The resulting fragment ions were scanned out in the low-pressure ion trap at normal scan rate, and recorded with the secondary electron multipliers. One MS/MS spectrum of a precursor mass was allowed before dynamic exclusion for 40 s. Lock-mass internal calibration was not enabled.

The spray and ion-source parameters were as follows. Ion spray voltage = 1800 V, no sheath and auxiliary gas flow, and capillary temperature of 260 °C.

### 4.5. Statistical and Bioinformatical Analyses

The LC-MS raw files were searched in MaxQuant (version 1.6.1.0; Max Planc Institute for Bichemistry, Martinsread, Germany), against the concatenated forward and reversed-decoy Swiss-Prot Homo sapiens database version downloaded 5 September 2019 and an implemented contaminant database by the Andromeda search engine [99,100]. Oxidation (M), Acetyl (Protein N-term) and Gln->pyro-Glu were set as variable modification, and carbamidomethylation (C) was set as fixed modification. Default settings were used, except that the alignment window was set to 1.4 min, match-between-runs enabled, ITMS/MS was set to 0.6 Da and LFQ count was set to 1.

Perseus (version 1.6.1.1; Max Planc Institute for Biochemistry) was used to process and filter the MaxQuant results [101]. Hierarchical clustering was performed in Perseus with Pearson correlations as distance metrics and complete linkage. GO analyses were performed using a GO tool [102] and filter hierarchy option was enabled for the biological processes (BP), molecular function (MF) and cellular compartment (CC) results to reduce the excess of results to a more concise subset, and GO terms with FDR < 0.05 were considered significantly enriched. Welch’s *t*-test and Z-statistics [103] were used to find significantly different fold change (FC) for proteins between different groups. Significantly regulated proteins were imported to the STRING database (version 11.0) using experiments and databases as interaction sources and 0.7 as minimum required interaction score (i.e., high confidence) [104]. Cytoscape (version 3.3.0; National Institute of General Medical Sciences, Bethesda, MD, USA) and ClusterONE (version 1.0; Exxact, Freemont, CA, USA) were used to visualize and classify protein networks of high cohesiveness, respectively [105,106].

## 5. Conclusions

Normal human MSCs and osteoblasts show constitutive extracellular release of a wide range of proteins, and there is a considerable overlap between the two cell types with regard to their protein release profiles. However, MSCs and osteoblasts differ especially with respect to their release of collagens as well as certain other extracellular matrix proteins. Both cell types release several proteins that support the growth of normal hematopoietic stem cells as well as hematopoietic and non-hematopoietic malignant cells. 

## Figures and Tables

**Figure 1 cancers-13-00062-f001:**
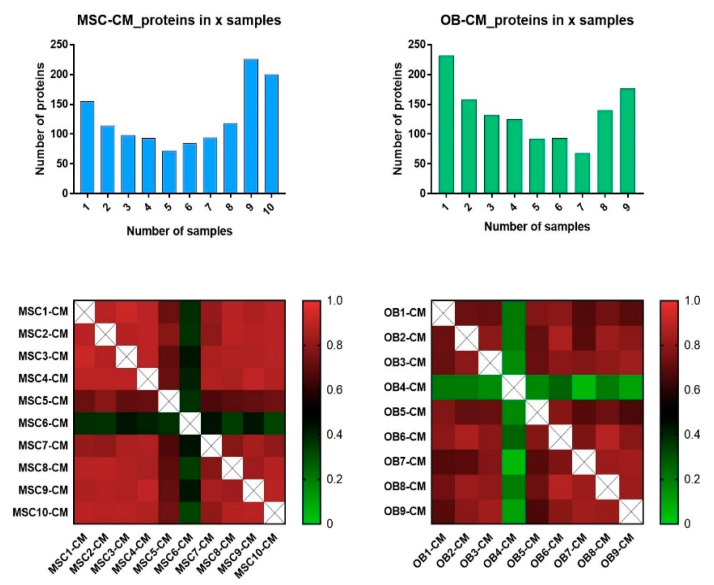
The constitutive protein release by normal MSCs and osteoblasts; a qualitative proteomic comparison of the various MSCs (10 donors) and osteoblast donors (OB, nine donors). Cells were cultured in serum-free medium for 48 h before supernatants were harvested and the protein levels analyzed in the culture supernatants. The upper two bar plots show the number of proteins (*y*-axis) that could be quantified for only one donor and up to all ten/nine donors/samples, (*x*-axis) for the MSCs (**upper left**) and osteoblasts (**upper right**). The lower part of the figure shows Pearson correlation plots based on the proteins that could be quantified for at least five donors/samples when analyzing the 10 MSC donors (**lower left**) and nine osteoblast donors (**lower right**).

**Figure 2 cancers-13-00062-f002:**
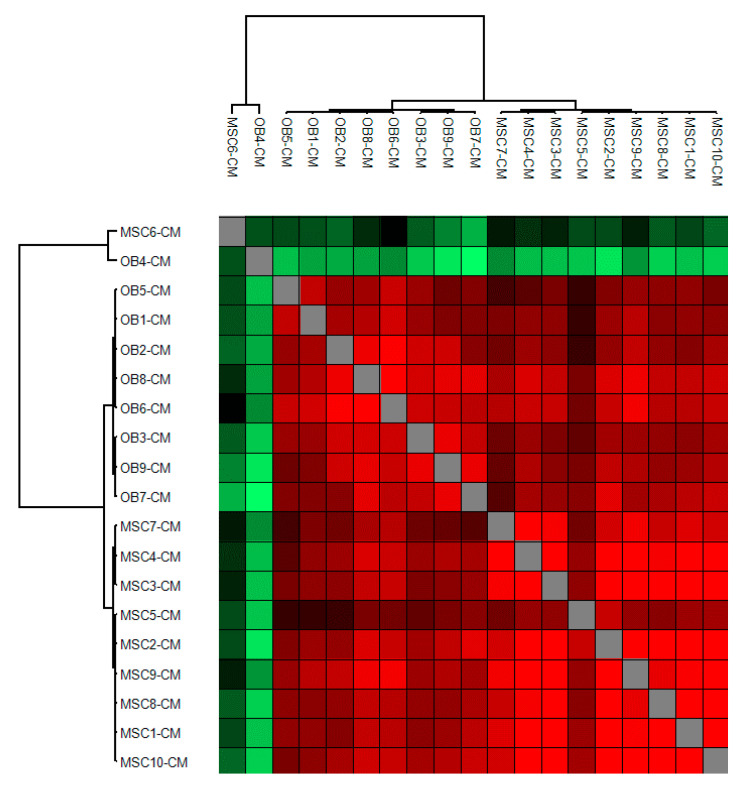
The constitutive protein release by normal MSCs and osteoblasts; hierarchical clustering of Pearson correlation values for the 10 MSC and the nine osteoblast donors. The cluster was generated using Pearson correlation as distance measure and complete linkage of the protein matrix with ≥4 valid values for at least one cell type (i.e., either MSCs or osteoblasts). We identified two main clusters corresponding to the two cell types and in addition two outliers (MSC6 and OB4).

**Figure 3 cancers-13-00062-f003:**
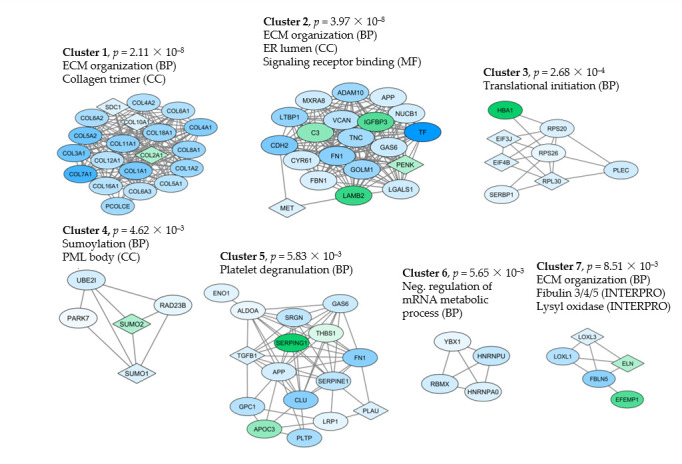
Network identification based on proteins that differed significantly when comparing the constitutive release by MSCs and osteoblasts (i.e., showing statistically significant differences or being exclusively released by only one cell type, see Section 2.5). The proteins were imported into the String database to identify interacting proteins, and the String network was imported to Cytoscape where the ClusterOne application was used to find protein clusters with high cohesiveness (Clusters 1–7). Proteins with higher abundance in MSCs are shown in blue, while green indicates higher abundance in osteoblasts (indicated to the lower left in the figure). GO terms were identified by the String database (BP, biological processes; CC, cellular compartments; MF, molecular functions; square symbols indicate that the protein is classified as unique for the indicated cell type).

**Figure 4 cancers-13-00062-f004:**
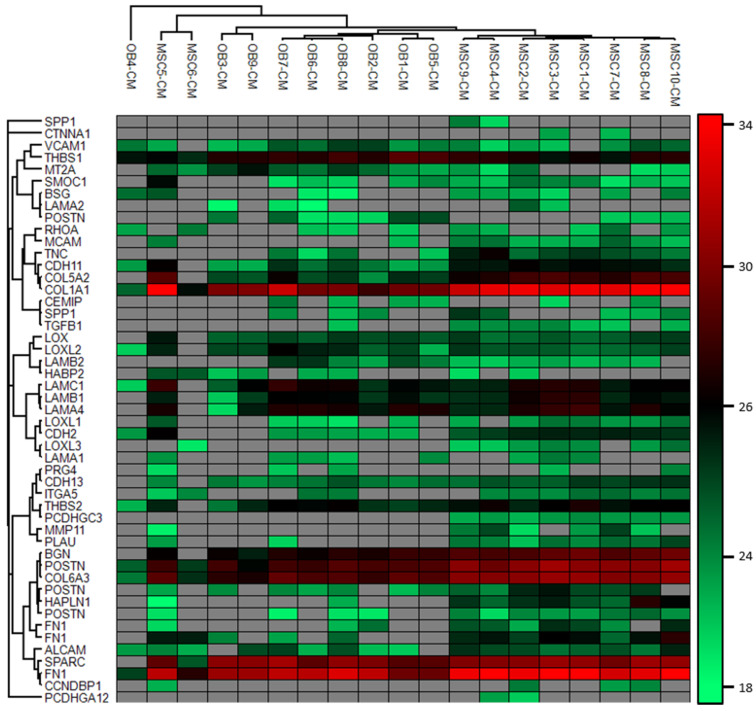
Hierarchical clustering analysis of proteins known to regulated bone marrow stem cell niches [1,2,3,4,5]. The analysis shows donor and cell type variation, i.e., protein intensity and detection of the proteins. Grey fields indicate that the protein was not detected in the donor conditioned medium. Most MSCs and OBs cluster together, but OB4, MSC5 and MSC6 differ from the others. In general, the most abundant proteins are THBS1, COL1A1, BGN, POSTN, COL6A3, SPARC and FN1, and particularly COL1A1, COL6A3 and POSTN1 are more abundant in MSCs. The cluster was generated using Pearson correlation as distance metrics and complete linkage of the protein matrix. The protein intensity is given as log2 LFQ intensity, and red indicates higher protein abundance. FN1, SPP1 and POSTN were detected as several isoforms and therefore represented several times.

**Table 1 cancers-13-00062-t001:** The characteristics of the MSC and osteoblast donors.

MSC	Gender	Age (Years)	Lot Number	Osteoblasts	Gender	Age (Years)	Lot Number
MSC Donor 1	Male	47	402Z027	hOB Donor 1	Male	54	422Z050
MSC Donor 2	Male	62	413Z021.4	hOB Donor 2	Male	58	422Z047.2
MSC Donor 3	Male	63	411Z011.4	hOB Donor 3	Male	63	415Z007.2
MSC Donor 4	Male	65	429Z013.1	hOB Donor 4	Male	58	422Z051
MSC Donor 5	Male	65	4031804.5	hOB Donor 5	Male	64	427Z036
MSC Donor 6	Male	70	429Z022	hOB Donor 6	Female	64	422Z031.2
MSC Donor 7	Female	44	402Z023	hOB Donor 7	Female	71	443Z004.2
MSC Donor 8	Female	57	409Z018.1	hOB Donor 8	Female	62	413Z026.2
MSC Donor 9	Female	66	421Z029.3	hOB Donor 9	Female	56	415Z011
MSC Donor 10	Female	73	427Z010.1				

Abbreviations: MSC, mesenchymal stem cells; hOB, human osteoblasts.

**Table 2 cancers-13-00062-t002:** The diversity of extracellular protein release by MSCs and osteoblasts during in vitro culture; a summary of protein classes showing at least five identified members. For more details see Appendix A.

Extracellular Matrix *(n = 34)* *Collagens (n = 22):* Twenty-one distinct collagen alpha-1 chains and the procollagen galactosyltransferase 1 enzyme; including Collagen alpha-1 (XVIII) chain that forms the antiangiogenic Endostatin protein through proteolytic cleavage. Both fibril-forming and fibril-associated collagens are included.
*Other extracellular matrix molecules (n = 12):* Including four fibulins, two fibrillins and six laminin subunits
Complement Factors *(n = 11)* Including factors contributing to the activation of the classical and alternative pathway, formation of the C1 complex and formation of the cell membrane attacking complex.
Other Proteases and Protease Inhibitors *(n = 26)* Including seven cathepsins, seven ADAM/ADAMTS proteases and 12 inhibitory serpins
Cell Surface Molecules *(n = 12)* Three cadherins, three HLA class I molecules plus HLA-DR alpha, five integrins.
Insulin-Like Growth Factor *(n = 9)* Including IGF II and eight IGF binding molecules or receptor components.
GTP associated proteins *(n = 21)* Five GTPases, five septins and 11 RAS-related proteins.
Protein Synthesis, Stabilization and Modulation *(n = 156)* Including five RNA binding proteins, 12 proteins involved in splicing, 44 ribosomal proteins, 21 proteins involved in translation initiation and elongation. In addition to these proteins:
*Chaperons:* Seven T complex proteins, 17 heat shock proteins and six Peptidyl-prolyl cis-trans isomerases.
*Posttranscriptional modulation:* four proteins involved in glycosylation, eight proteins involved in protein phosphorylation, 12 proteins involved in ubiquitination together with 20 proteasomal proteins.
Cytoskeleton *(n = 22)* 19 actin and actin-interacting proteins, together with three tropomyosins.
Organellar Functions/V-Atpase *(n = 9)*
Nuclear Proteins *(n = 49)* Including 18 histones, 20 non-histone chromatin proteins and 11 other nucleoproteins.
Intracellular Transport *(n = 14)* Six 14-3-3 proteins and eight solute carrier transporters
Metabolism-Redox Balance *(n = 10)* Four lipoproteins and six peroxiredoxins.
S100 Proteins *(n = 8)*

**Table 3 cancers-13-00062-t003:** An overview of released proteins showing significant differences between in vitro cultured MSC and osteoblasts. For more detailed information see Appendix A).

Increased in MSCs *(n = 82)*	Increased in Osteoblasts *(n = 36)*
**Extracellular Matrix**	
Collagens: COL1A1, COL1A2, COOL3A1, COL4A1, COL5A2, COL6A2, COL7A1, COL8A1, COL10A1, COL11A1, COL18A1 Others: ABI3BP, ECM1, EMILIN1, FBLN5, HAPLN1, HMCN1, NID1, POSTN (stem cell maintenance), TNC	Collagens: COL2A1, Others: ACAN, EFEMp1, ELN, MXRA5
**Proteases, Protease Inhibitors**	
Proteases: ADAM9, ADAM10, AEBP1, CTSC, FN1, MMP1, MMP11, PCOLCE (collagen), PLAU (coagulation) Inhibitors: SERPINB7	Proteases: CTSA, CTSD Inhibtor: SERPING1
**Other Enzymes**	
ENPP1(phosphodiesterase), LOXL3 (lysyl oxidase), MAN1A1 (mannosidase), PTK7 (tyrosine kinase), SRPX (antioxidant)	C3 (complement), F10 (coagulation)
**Protein Modulation/Stabilization**	
CLU (chaperone), CTSC (chaperone), GRPEL1 (chaperone/HSP70)	HSPA6 (chaperon), SUMO2
**Extracellular Mediators**	
CHID1 (growth factor, chemokine interaction), CLEC11A (growth factor), GREM1 (BMP antagonist), LTBP1 (TGF), MET (HGF receptor), TGFB1 (cytokine)	IGFALS, IGFBP3, NBL1 (BMP antagonist), PENK
**Cell–matrix Adhesion**	
ADAM10, CSPG4 (chondroitin), FN1, SDC1, TNC, VASP	SDCBP, THBS4
**Cell–cell Contact**	
CCDC80 (cadherin), CDH11 (cadherin), CDH2 (cadherin), CHID1 (syndecan), HMCN1 (desmosomes), MCAM, PCDHGC3 (cadherin)	THBS4
**Cell Surface Molecules**	
CD109, CDH11 (cadherin), CDH2 (cadherin), CHST11 (chondroitin), CSPG4 (chondroitin), NRP2 (semaphorin), SDC1, SEMA7A	VNN1
**Intracellular Transport, Endocytosis, Secretion**	
ATP6AP1 (organellar acidification, V-ATPase), CHST11 (Golgi), CTSC (lysosome), GOLM1 (ER/Golgi), HSPA9 (ER/Golgi), MVP (nucleo-cytoplasmic), RRBP1 (ER), VAPA	RAB7A, SDCBP
**Cytoskeleton**	
ARPC3, MAPRE1, VAPA, VASP	MYH14, SDCBP
**Protein Synthesis, Ribosome**	
EIFRJ, EIF4B, RPL30, RRBP1	PSMB8 (proteasome)
**Intracellular Signaling**	
CD109 (TGF), LTBP1 (TGF), MET (HGF receptor), MVP, PTK7 (Wnt)	CNPY2, PODN (Wnt), RAB7A (GTPase), SCUBE3 (TGFβ), RCSD1 (Wnt)
**Angiogenesis**	
CTHRC1, ECM1, FBLN5, NRP2 (VEGF)	
**Metabolism**	
ALDOC (glycolysis), GATD3A (mitochondria), GRPEL1 (mitochondria), HSPA9 (mitochondria), LDLR (lysosome, cholesterol), LIPA (lysosome, cholesterol), OCT1, mitochondria, ketones), PLTP (cholesterol), SDC1, TF (iron)	AHCY (adenosine, homocystein), APOC3 (lipid)
**Transcription, RNA**	
AEBP1, HNRNPUL2 (noncoding RNA), HSBP1	PTMS, TSSK4
**Others**	
ABI3BP (senescence, tumor suppressor), APEX1 (DNA repair, endonuclease), SRPX (antioxidant enzyme), SUMO1 (protein modulation)	BAX (apoptosis), HBa1 (hemoglobin), MT2 (antioxidant), SEPP (antioxidant)

Abbreviations: BMP, bone morphogenic protein; ECM, extracellular matrix; ER, endoplasmatic reticulum; HGF, hepatocyte growth factor; TGF, transforming growth factor; VEGF, vascular endothelial growth factor; Wnt pathway, Wingless and Int-1 pathway.

**Table 4 cancers-13-00062-t004:** Proteins showing qualitative differences when comparing MSCs and osteoblasts. For each of the two cell types, the table shows the number of donors with detectable levels (indicated to the left in each of the two main columns) together with the name of the protein.

Detected Only in MSC (≥7 Out of 10 Donors)		Detected Only in Osteoblasts (≥6 Out of 9 Donors)	
9	Stanniocalcin-1	8	Aggrecan core protein 2
9	Collagen alpha-1(X) chain	7	Coagulation factor X
9	Mannosyl-oligosaccharide 1,2-alpha-mannosidase IA	7 7	Selenoprotein P CapZ-interacting protein
9	60S ribosomal protein L30	6	Proenkephalin-A;
8	Lysyl oxidase homolog 3	6	Collagen alpha-1(II) chain
8	Chitinase domain-containing protein 1		
8	Carbohydrate sulfotransferase 11		
8	Protocadherin gamma-C3		
8	Hepatocyte growth factor receptor		
8	Microtubule-associated protein RP/EB family member 1		
7	Serpin B7		
7	Low-density lipoprotein receptor		
7	Stromelysin-3		
7	Dipeptidyl peptidase 1;Dipeptidyl peptidase 1 exclusion domain chain; Dipeptidyl peptidase 1 heavy chain; Dipeptidyl peptidase 1 light chain		
7	Lysosomal acid lipase/cholesteryl ester hydrolase		
7	Glutamine amidotransferase-like class 1 domain-containing protein 3A, mitochondrial		

**Table 5 cancers-13-00062-t005:** Proteins important for the support of hematopoietic stem cells in the bone marrow stem cell niches [1,2,3,4,5]; a summary of relevant proteins also detected in our present proteomic study of the MSC and osteoblast secretome. Proteins being important for the molecular interactions between stem cells and normal bone marrow stromal cells or extracellular matrix molecules in the bone marrow stem cell niches were identified based on the five previous review articles. The table lists all identified proteins that in addition were detected in our present proteomic study of MSC/osteoblast secretoma.

**Extracellular Matrix**	
Collagen	Collagen alpha-1(I) chain (COL1A1), Collagen alpha-2(V) chain (COL5A2), Collagen alpha-3(VI) chain (COL6A3)
Laminin	Laminin subunit alpha-1 (LAMA1), Laminin subunit alpha-2 (LAMA2), Laminin subunit alpha-4 (LAMA4), Laminin subunit beta-1 (LAMB1), Laminin subunit beta-2 (LAMB2), Laminin subunit gamma-1 (LAMC1)
Others	Fibronectin (FN1), Tenascin (TNC), secreted protein acidic and cysteine rich (SPARC), SPARC-related modular calcium-binding protein 1 (SMOC1)
**Proteases**	
	Protein-lysine 6-oxidase (LOX), Lysyl oxidase homolog 1 (LOXL1), Lysyl oxidase homolog 2 (LOXL2), Lysyl oxidase homolog 3 (LOXL3), Stromelysin-3 (MMP11) Urokinase-type plasminogen activator (PLAU)
**Cell Surface Molecules**	
Cadherin	Cadherin-11 (CDH11), Cadherin-13 (CDH13), Cadherin-2 (CDH2), Catenin alpha-1 (CTNNA1), Protocadherin gamma-A12 (PCDHGA12), Protocadherin gamma-C3 (PCDHGC3)
Hyaluran	Hyaluronan-binding protein 2 (HABP2), Hyaluronan and proteoglycan link protein 1 (HAPLN1), Cell migration-inducing and hyaluronan-binding protein (CEMIC)
Thrombospondin	Thrombospondin-1 (THBS1), Thrombospondin-2 (THBS2)
Proteoglycan	Biglycan (BGN), Proteoglycan 4; Proteoglycan 4 C-terminal part (PTG4)
Others	Vascular cell adhesion protein 1 (VCAM1), Integrin alpha-5 (ITGA5), Cell surface glycoprotein MUC18 (MCAM), CD166 (ALCAM), Basigin (BSG)
**Soluble Mediators**	
	Osteopontin (OPN), Transforming growth factor beta-1; Latency-associated peptide (TGFB1), Periostin (POSTN)
**Intracellular Regulators**	
	Cyclin-D1-binding protein 1 (CCNDBP1) Metallothionein 1 (MT1), Metallothionein 2 (MT2), Ras-related C3 botulinum toxin substrate 1 (RAC1), Transforming protein RhoA (RHOA)

## Data Availability

The data presented in this study are available on request from the corresponding author.

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
