# Peer review of "The Extracellular Bone Marrow Microenvironment—A Proteomic Comparison of Constitutive Protein Release by In Vitro Cultured Osteoblasts and Mesenchymal Stem Cells"

_cancers, 2020, doi:10.3390/cancers13010062_

Round 1
Reviewer 1 Report
General
Manuscript per se is well and fluently written, and contains relevant scientific data, i.e., secreted proteome of in vitro cultured, important cell subtypes of bone marrow stroma. And proteomics is a highly specific, expertise- and labour-demanding technique. Also number of biological replicates is high.
However, considering the journal and special issue "Microenvironment and Cancer Progression", the relevance of the findings to bone marrow cancer progression remains rather weak. Merely reporting protein lists secreted by normal cells leaves a question that how would these cells support (or prevent?) cancer progression. I’m not sure if this is the best forum for this manuscript. In present form, it remains disconnected from the scope.
I suggest some additional experimentation or data analysis related to cancer (or bone marrow niche). Or presentation of data a bit differently. I miss more clear presentation of which proteins found in these two cell types (or either cell type) could be relevant to cancer and which part of cancer progression. Maybe a figure illustrating this?
For example of rows 322-327, there’s a list of proteins relevant for cancer, but were they found on this data? Not? The next chapter describes some relevant proteins for different cancer types and comparison to this data. Still the relevance remains a bit weak. Is there really nothing known about hematopoietic cancers, especially leukemia? Then that should be emphasized.
I was surprised that the group has published a paper about MSCs supporting AML cells but they only refer to that paper where they mention that MSCs “release a wide range of soluble mediators” and not mention that they support AML survival and proliferation (Brenner, A.K.; Nepstad, I.; Bruserud, Ø. Mesenchymal Stem Cells Support Survival and Proliferation of 556 Primary Human Acute Myeloid Leukemia Cells through Heterogeneous Molecular Mechanisms. Front 557 Immunol 2017, 8, 106 ). This is the kind of data I would like to see in this paper to get more relevance to cancer. In that paper, did the group study the role of MSC secretome/proteome in supporting AML?
I understand that it is not an easy or minor task, but could it be shown that proteome supports the survival of hematopoietic cancer cells?
Minor comments:
Introduction/discussion:
Authors could include some information about the frequency of MSC vs osteoblasts in normal bone marrow and is this changed in cancer, or is it known? What is known about the roles of these two cell types, or differences in their roles in BM?
How about comparison of this data to proteomic data that has been published on MSCs and osteoblasts. Just to see if there is overlap or not? Should be compared and mentioned, at least. There might be some contaminating membrane or cytosolic proteins? Or of course they can be present in exosomes/extracellular vesicles. There is some published data on MSC vesicles, comparison to that would be important as well.
Methods:
- Does this medium contain animal-derived components, that might influence the proteome?
- I could not find IMDM medium from promocell
- Was the condition and viability of cells checked after 48h starvation before medium collection? That there were no apoptotic cells, attached cells, cell membrane particles? This should be commented
- Some comment of these culture conditions, how they relate to live bone marrow conditions, in discussion? e.g. oxygen levels? in vitro vs in vivo cells?
Table 5 heading is a bit confusing, explain more clearly. Proteins common in MSC and osteoblast proteome? Also word “secretoma” is confusing.
Reviewer 2 Report
The authors characterized, by using a proteomic approach, the protein release by in vitro cultured human MSCs and osteoblast from healthy donors. the study is clear and well organized. However, it is restricted to normal cells. It could be interesting to use the same approach on MSCs and osteoblasts derived from subjects affected by cancers highly dependent by microenvironment. Moreover, it could be useful to study how the protein release changes after co-culturing MSC/osteoblast with tumor cells.
Reviewer 3 Report
cancers-1038737-peer-review-v1
General Comments: Overall, very well written and presented study. A few general comments:
1) While there is identification of secreted proteins that are involved in cancer pathogenesis, there is no specific interrogation of the secretome of MSCs and osteoblasts exposed to cancer related factors or MSCs and osteoblasts derived from cancer patients.
2) More emphasis should be placed on the fact MSCs are precursors of osteoblasts.
3) In the abstract, there is no discussion of the relevance of the findings in how it relates to cancer.
4) MSC6, which behaved different biologically in many ways, took a long time to get to confluency for CM preparation. This was the oldest male donor. Could there be something wrong with these cells such that this factor could warrant their removal from the study? For example, mycoplasma infection, etc. Was the cell viability assessed?
5) Do differences exist between male and female cells?
6) While a nice proteomic approach, and many proteins identified, there is no validation of protein levels by ELISA (or cellular mRNA expression).
Round 2
Reviewer 1 Report
The authors have adequately answered all my questions and concerns and rewritten the text accordingly. Thus manuscript has considerably improved and is more relevant to the cancer context in the present form.
Reviewer 3 Report
The authors have nicely addressed all of my comments.